# Should We Expect an Increase in the Number of Cancer Cases in People with Long COVID?

**DOI:** 10.3390/microorganisms11030713

**Published:** 2023-03-09

**Authors:** Ana Amiama-Roig, Laura Pérez-Martínez, Pilar Rodríguez Ledo, Eva M. Verdugo-Sivianes, José-Ramón Blanco

**Affiliations:** 1Centro de Investigación Biomédica de La Rioja (CIBIR), 26006 Logroño, Spain; 2Hospital Universitario Lucus Augusti, 27003 Lugo, Spain; 3Instituto de Biomedicina de Sevilla (IBIS), Hospital Universitario Virgen del Rocío (HUVR), Consejo Superior de Investigaciones Científicas, Universidad de Sevilla, 41013 Seville, Spain; 4CIBERONC, Instituto de Salud Carlos III, 28029 Madrid, Spain; 5Servicio de Enfermedades Infecciosas, Hospital Universitario San Pedro, 26006 Logroño, Spain

**Keywords:** cancer, COVID-19, immunosuppression, inflammation, senescence, SARS-CoV-2

## Abstract

The relationship between viral infections and the risk of developing cancer is well known. Multiple mechanisms participate in and determine this process. The COVID-19 pandemic caused by the SARS-CoV-2 virus has resulted in the deaths of millions of people worldwide. Although the effects of COVID-19 are limited for most people, a large number of people continue to show symptoms for a long period of time (long COVID). Several studies have suggested that cancer could also be a potential long-term complication of the virus; however, the causes of this risk are not yet well understood. In this review, we investigated arguments that could support or reject this possibility.

## 1. Introduction

The link between viral infections and the risk of developing cancer is well known. It is estimated that 15.4% of all cancer cases can be attributed to carcinogenic infections, for which viruses are the main risk factor [1]. In 1964, Epstein et al. [2] identified the first human oncogenic virus. Since then, multiple viruses have been studied for their potential role in aiding the development of cancer [3]. To date, at least seven human cancer oncogenic viruses have been shown to have strong connections to various forms of cancer in humans, including the Epstein–Barr virus, human papillomavirus and the hepatitis B and C viruses [4]. The mechanisms involved are varied and range from chronic inflammation to immunosuppression and DNA modification. Indeed, viruses can transform cells via a variety of mechanisms, such as providing external oncogenes, over-activating human oncogenes and/or inhibiting tumour suppressors [5].

Recently, severe acute respiratory syndrome coronavirus 2 (SARS-CoV-2) and the ongoing outbreak of the novel coronavirus disease 2019 (also known as COVID-19) that started in Wuhan (China) have rapidly become the most important global health problems, infecting more than 600 million people and causing more than 6.5 million deaths worldwide [6]. Although COVID-19 was clearly associated with severe respiratory disease at the beginning of the pandemic [7], it was soon realised that COVID-19 is a systemic infection implicated in multiple extra-pulmonary complications (i.e., myocardial injury and neurological disease) [8,9]. In addition to the current clinical situation, there are also increasing concerns about the long-term consequences of these infections. One such concern is the increased risk of new chronic diseases (e.g., diabetes) [10,11]. Indeed, a significant number of COVID-19 patients (37–57%) [12,13] have reported a wide range of persistent symptoms (i.e., fatigue or muscle weakness, general pain, weight loss, etc.) in the 6 months after infection, known as long COVID (LC) [14]. Merad et al. [15] suggested that the leading hypotheses include the persistence of viral antigens and RNA [16] in tissues that drive chronic inflammation, autoimmunity [17,18] that is triggered by acute viral infection, dysbiosis in the microbiome or virome and the possibility of chronic tissue damage in different organs and tissues [19,20].

The structural proteins in SARS-CoV-2 include the spike glycoprotein (which is known for its pathogenicity, i.e., it facilitates virus entry into healthy cells), the nucleocapsid (which is implicated in genome replication), a membrane protein (which is implicated in virus assembly) and the envelope protein. SARS-CoV-2 binds to the angiotensin-converting enzyme 2 (ACE2), which is a cell surface receptor that is highly expressed in the ileum, vascular endothelia, kidneys, lungs, kidneys, testes and immune cells [21,22,23].

As we learn more about the long-term complications of this new disease, we should also be alert to the possibility of correlations with other longer-term complications, such as cancer. By assessing the different mechanisms involved in the development of cancer after COVID-19, we reviewed the different arguments that support or reject the possibility of a correlation between acute and long COVID and the risk of cancer. References for this review were identified through searches of PubMed (MeSH terms) with the terms “cancer”, “carcinogenesis”, “cellular senescence”, “coronavirus infections”, “COVID”, “DNA damage”, “genome, viral”, “humans”, “immune system”, “inflammation”, “long COVID”, “neoplasm”, “oncogene”, “oncovirus”, “oncogenic viruses”, “post-acute COVID-19 syndrome”, “post-acute sequelae of COVID-19”, “receptors, coronavirus”, “residual SARS-CoV-2”, “SARS-CoV-2”, “spike glycoprotein, coronavirus”, “spike protein”, and “virus integration”. 

English language articles from these searches and relevant references cited in those articles were reviewed.

## 2. Senescence and Viral Infections

Biological ageing is associated with higher risks of developing several types of cancer [24]. One of the most important mechanisms in the ageing process is referred to as cellular senescence, which is also implicated in many age-related comorbidities [25], including cancer. Cellular senescence (CS) is a program activated by normal cells in response to different types of stress, such as DNA damage, telomeres shortening, oxidative stress and oncogene stimuli. When cells enter in senescence, they leave the cell cycle, lose their capability to proliferate in response to mitogenic stimuli and undergo multiple phenotypic changes, including the increased activity of lysosomal acidic beta-galactosidase, among others. Therefore, cellular senescence could be considered as a tumour suppressive mechanism by itself. However, CS is considered to be a double-edged sword [26]. One of the most important features of senescent cells is the secretion of proinflammatory factors, which is known as the senescence-associated secretory phenotype (SASP). This phenotype involves the secretion of cytokines, chemokines, proteases, growth factors and extracellular media elements, as well as degrading enzymes. The accumulation of senescent cells has been suggested to be one of the causes of chronic inflammation that may eventually induce tumorigenesis [27,28,29,30]. 

Following a viral infection, a large number of cellular stress response pathways are activated that, in most cases, help to control virus replication. In this sense, numerous different articles have described the links between viral infections and CS [31,32,33]. Indeed, several viruses have developed mechanisms that interfere with cellular senescence, which could be interpreted as viral strategies to evade the cellular antiviral system [33]. Baz-Martinez et al. [31] explored the antiviral power of CS using the vesicular stomatitis virus as a model and evaluated different stimuli involved in cellular senescence in both murine and human primary and tumour cells. They observed that cellular senescence demonstrated antiviral activity that reduced virus replication and infectivity and viral protein synthesis and apoptosis, irrespective of the trigger [31]. 

There is a reciprocal relationship between COVID-19 and CS, since coronavirus infections promote CS. Camell et al. [34] reported that SARS-CoV-2 surface antigen spike-1 was able to amplify the SASP of cultured human senescent cells and that a related mouse β-coronavirus (HMV) increased proinflammatory SASP factors and senescent cell burden in infected mice. Tripathi et al. [35] also observed that SARS-CoV-2 induced cellular senescence in human non-senescent cells and exacerbated SASP through the toll-like receptor (TLR)-3. In a similar way, Evangelou et al. [36] examined autopsied lung tissue samples from COVID-19 patients and age-matched non-COVID-19 controls and found that infected cells exhibited cellular senescence and a proinflammatory phenotype. They suggested that cellular senescence was mediated by DNA damage and the activation of the DNA damage response pathway [36]. Recently, Balnis et al. [37] published the first reported evidence that DNA methylation changes in circulating leukocytes endured for at least 1 year after recovery from acute COVID-19 infection.

Along the same line, Moneglli et al. [38] analysed a group of 144 age- and sex-matched COVID-19-free persons with some risk factors and post-COVID-19 patients. In the post-COVID-19 patients, an acceleration of their biological blood clock was observed, particularly among those under 60 years of age [38]. Significant telomere shortening, which is another marker of ageing, was also observed. Cao et al. [39] found that COVID-19 could accelerate epigenetic ageing in infected patients, although this process could be reversed in some patients. Although no patients with LC were included in that study, the authors speculated that epigenetic ageing and telomere length could contribute to LC symptoms. Victor et al. [40] observed that SARS-CoV-2 infection resulted in transcriptional upregulation of the DNA damage response to ataxia telangiectasia and Rad3-related proteins, as well as a reduction in telomere length [40]. These conditions have been associated with genome instability but so far, the clinical implications are unknown. However, Franzen et al. [41] analysed blood samples from 50 patients who were hospitalised with severe SARS-CoV-2 infections and did not observe any evidence of accelerated epigenetic ageing or significantly shortened telomere lengths.

## 3. Chronic Inflammation and Viral Infections

Chronic inflammation has been identified as an important step in tumorigenesis [42] and two particular events appear to be required to activate that step [43]. Firstly, tumour-associated viruses must develop mechanisms that help them to evade host immune systems. Secondly, persistent infections must be capable of inducing mild but persistent inflammation. Indeed, the transformation of cells by viruses is important in the development of tumorigenesis. Chronic inflammation may also increase the generation of mutations and may consequently increase the risk of tumour development [44].

There is evidence to support the idea that inflammatory pathways remain altered for long periods of time after COVID-19 infection. Doykov et al. [45] observed that 40–60 days after COVID-19 infection, a significant inflammatory response was observed that was associated with an anti-inflammatory response (which is characterised by a Th17 inflammatory profile), a reduced anti-inflammatory response (which is characterised by lower levels of IL-10 and IL-4) and mitochondrial stress, even in asymptomatic patients. Convalescent patients with mild or asymptomatic infections have also shown neutrophil dysfunction, which could increase a patient’s susceptibility to cancer [46]. This is caused by an increase in the count of low-density neutrophils (LDNs), whose immunosuppressive activity is well known. The increase in LDNs correlates with a poor T-cell response and greater disease severity. It has been hypothesised that lymphopenia could lead to the inability to control the infection or viral dissemination [46]. So, an increase in myeloid-derived suppressor cells could diminish the inflammatory response and inhibit effector T-cell response and IFN-γ production. In the same way, Queiroz et al. [47] observed significantly higher levels of IL-17 and IL-2 in subjects with LC. However, the cytokine levels of IFN-γ, TNF and IL-6 did not show any significant differences. No change in IL-6 is important because in cases where there is the hyperactivation of this interleukin, the IL-6/JAK/STAT3 pathway is also hyperactivated, which occurs in many types of cancer [48].

Another possibility could be the “reactivation” of SARS-CoV-2 or other viruses. It is believed that residual virus cells could remain in certain organs or tissues, which could result in long-lasting immunomodulatory effects. This could explain the low-grade inflammation that has been described in some convalescent patients [49,50]. This chronic inflammation coupled with oxidative stress could lead to tissue and DNA damage. TLR activation is induced in response to RNA viruses, thereby stimulating the synthesis of proinflammatory cytokines and interferons [51] that contribute to limiting viral infection or viral replication. However, coronaviruses can antagonise interferons, thus evading host immune systems [52].

Finally, the SARS-CoV-2 spike protein contains a furin-like cleavage site that is absent in the other SARS-like CoVs, so its inhibition may represent a potential antiviral strategy [53]. This spike protein promotes the activation of the NLRP3 inflammasome and NF-κB inflammatory pathways [54,55,56]. Elevated inflammasome pathways, which are present in older people, have been associated with age-related comorbidities [57,58]. Indeed, increased inflammasome activity as a consequence of a viral infection may contribute to the age-related impairment of immune responses.

## 4. Chronic Virus Infection, Residual Virus Proteins and Cancer Risk

Many RNA viruses can cause persistent infections [59] and some have been implicated in higher cancer risks. A good example is the hepatitis C virus, which has been implicated in the development of liver cancer. This relationship is well established and the risk is closely linked to the duration of hepatitis C infection. Indeed, at a molecular level, hepatitis C proteins (e.g., NS3, NSA4B, etc.) have oncogenic potential [60].

The oncogenic mechanisms of most viruses involve the continued expression of specific viral gene products that regulate proliferative or anti-apoptotic activity through interactions with cellular gene products. However, some viruses, such as HIV, exert oncogenic effects through indirect mechanisms, e.g., immunosuppression [61].

Cheung et al. [62] detected residual SARS-CoV-2 nucleocapsid proteins in multiple extrapulmonary tissue samples (i.e., tissues from the colon, appendix, ileum, liver and lymph nodes) from patients who had recovered from COVID-19 for up to 6 months after testing negative for SARS-CoV-2. However, they were unable to detect viral RNA in some of the samples. They also observed that SARS-CoV-2-specific memory T cells could be maintained in both blood and tissues over long periods of time [62].

Natarajan et al. [63] reported that while there was no excretion of SARS-CoV-2 RNA in the oropharynx 4 months after infection, 12.7% of patients were still excreting SARS-CoV-2 RNA in their faeces 4 months after infection and 3.8% were still excreting it after 7 months; however, this does not necessarily indicate the presence of a live virus. Unfortunately, the samples were not cultured.

Zhuo et al. [64] also observed that the microbiota of patients with acute COVID-19 infections were altered. This alteration was characterised by a depletion of beneficial bacteria (commensals) and a proliferation of opportunistic pathogens in the gut. They found that this state persisted once the SARS-CoV-2 infection was cleared from the respiratory tract and some respiratory fungal pathogens, such as *Aspergilus flavus* and *A. niger*, were found in faecal samples.

It has been suggested that another potential reservoir could be the genital tract [65]. So, electronic microscopy was used to analyse penile tissue from two patients who had recovered from COVID-19 and showed the presence of coronavirus-like spike proteins, although spike protein-positive cells were not detected by immunofluorescence, despite the patients producing positive COVID-19 PCR tests [66]. Meanwhile, other studies have not been able to detect this virus in semen samples of COVID-19 patients after 1 month of COVID-19 infection [67].

Zollner et al. [68] performed endoscopies on 46 patients with inflammatory bowel disease who also had acute COVID-19 infections (confirmed by PCR testing) nearly 7 months before. They found antigen persistence in 52–70% of the patients. Viral nucleocapsid proteins persisted in the gut epithelium and CD8+ T cells of 52% of the patients. However, the expression of SARS-CoV-2 antigens was not detectable in the stool samples and viral antigen persistence was unrelated to the severity of the COVID-19 infection, immunosuppressive therapy and gut inflammation [68]. LC symptoms were reported by the majority of patients with viral antigen persistence, but not by patients without viral antigen persistence, suggesting that this could be the basis for LC complications. The authors were unable to culture SARS-CoV-2 using gut tissue samples from patients with viral antigen persistence.

Goh et al. [69] also reported two cases of LC with persistent viral antigens and/or RNA. The first case had residual virus cells in both gastrointestinal and non-gastrointestinal tissue for up to 426 days, while the second case had residual virus cells in non-gastrointestinal tissue. Unfortunately, fresh tissue and blood samples were not collected, so viral viability could not be assessed.

More recently, Bussani et al. [70] performed post-mortem analyses in 27 consecutive patients who had apparently recovered from COVID-19 (repeated viral negativity in nasopharyngeal swabs or bronchioalveolar lavage for 11–300 consecutive days), but had progressively worsened in their clinical conditions. However, despite the apparent molecular negativity, these patients still harboured virus-infected cells in their lungs. According to the same authors, the role of persistent infection in the pathogenesis of long COVID remains to be established.

Indirect evidence has been reported that supports the viral reservoir hypothesis. So, Peluso et al. [71] collected data about the potential benefits of treating LC patients with nirmatrelvir/ritonavir, which is a protease inhibitor with demonstrated activity against SARS-CoV-2 [72]. This could support the hypothesis that persistent viral activity, particularly in the tissues, could be implicated in LC [71].

The potential integration of SARS-CoV-2 into the human genome has been also evaluated. So far, multiple researchers have refuted this possibility [73,74,75]. However, Zhang et al. [76] found that SARS-CoV-2 sequences could be reverse-transcribed and integrated into the DNA of infected cultured human cells. However, in response to this article, some authors have criticised the experimental design of the study and the interpretation of the results [74,75]. Briggs et al. [73] also suggested that theSARS-CoV-2 integration was likely to be artificial, stemming from amplicon DNA contamination and/or other unintended processes. These are undoubtedly relevant aspects that need to be studied in more detail.

Craddock et al. [77] analysed the levels of circulating SARS-CoV-2 components, such as the spike protein and viral RNA, in patients who were hospitalised with acute COVID-19 and patients with and without LC. After comparing patients with and without LC symptoms, the authors found that the spike protein and viral RNA were at higher levels in patients with LC symptoms than in acute COVID-19 patients [76]. The authors also observed that the percentage positivity of circulating viral RNA was increased in LC patients compared to acute COVID-19 patients, while the spike protein positivity remained the same [77]. Swank et al. [78] conducted a retrospective study and also found the presence of circulating spike protein in patients with LC for up to 12 months after COVID-19 infection. However, the mechanisms of this situation are not well known; while some have suggested that a reservoir of active virus cells persists in the body [78], others have reported that the persistence of viral cells is probably not due to an actively replicating virus infection [77].

Meanwhile, other research has described the overlap between SARS-CoV-2 spike and tumour-suppressor proteins and has reported autoimmune cross-reactivity to be a potential mechanism underlying prospective cancer insurgence following exposure to SARS-CoV-2 [79]. This could be an interesting topic to investigate.

In addition, another notable aspect that is associated with chronic viral infections is accelerated ageing, as measured by the “epigenetic clock” [80], the implications of which were described previously [38].

## 5. The Oncogenic Potential of SARS-CoV-2

Another mechanism that could implicate SARS-CoV-2 in the risk of developing cancer could be the oncogenic potential. SARS-CoV-2 has developed similar strategies to other viruses (e.g., the Epstein–Barr virus) to control p53, which represents a threat to the virus [81,82]. Because the onco-suppressive protein p53 plays an important role in the apoptotic signalling pathway, it has been hypothesized that the long-term inhibition of p53 by SARS-CoV-2 could produce carcinogenic effects [83]. Gomez-Carballa et al. [83] examined three gene expression datasets and demonstrated that p53 was downregulated during acute SARS-CoV-2 infection and LC. The multidomain non-structural protein 3 (Nsp3) SARS-CoV-2 protein promotes the degradation of p53 through the activation of RING and E3 ubiquitin ligase, which are implicated in apoptosis. In addition, coronaviruses encode endoribonuclease non-structural protein 15 (Nsp15), which interacts with another important tumour suppressor, the retinoblastoma protein (pRb) [84], through the ubiquitin–proteasome pathway [85]. Nsp15 expression leads to a reduction in pRb expression, which induces cell transformation, chromosomal instability and changes in cell cycle-associated gene expression [86,87]. This is highly relevant because p53 and pRb are recognised as important tumour suppressor genes [84,88]. Given the ability of SARS-CoV-2 to inhibit both p53 and pRb, SARS-CoV-2 could have oncogenic potential. In fact, SARS-CoV-2 non-structural protein 1 can interact with DNA polymerase alpha (Pol-a), being a source of instability because Pol-a is not only involved in the initiation of replication, but also in the coordination of cell cycle progression and the DNA damage response. A high molecular mimicry has also been reported between the spike glycoprotein and various tumour suppressor proteins (e.g., BC11B, BRCA1 and 2, PLAT2 and 3, etc.) [79]. These repeated epitopes have also been found in multiple infectious pathogens, opening up the possibility of immunologic imprinting. This phenomenon could lead to autoimmune cross-reactivity and, potentially, cancer development.

Ebrahimi et al. [89] evaluated the possible correlations between SARS-CoV viruses and cancer in an in silico study model. Different analyses showed that four genes (PTEN (proliferation and cellular death), CREB1 (transcription activator), CASP3 (cell apoptosis) and SMAD3 (transcription factor and cell proliferation) were key in cancer development. According to the TCGA database results, these four genes were upregulated in pancreatic adenocarcinoma [89]. In a similar way, Zhao et al. [90] performed a genome-wide cross-trait analysis to investigate the shared genetic architectures and putative genetic associations between COVID-19 and the three main female-specific cancers (breast cancer, epithelial ovarian cancer and endometrial cancer). Although the authors did not find any evidence of genetic correlations between COVID-19 and the female-specific cancers, the cross-trait meta-analysis found that these conditions shared multiple mechanistic pathways (connecting the hematologic system, immune system and cell proliferation), especially in breast cancer and ovarian cancer [90].

Finally, Shen et al. [91], driven by the higher incidence of COVID-19 among cancer patients, used bioinformatics techniques to analyse the differentially expressed genes (DEGs) that are common to three of the most prevalent cancers (breast, liver and colon) and COVID-19. The authors identified 38 DEGs through a cross-comparison evaluation that was conducted on Jvenn. They also performed GO and KEGG enrichment analyses, starting from those 38 DEGs. They found that the DEGs were enriched in “elastic fibre assembly”, “collagen-containing extracellular matrices” and “oestrogen 2-hydoroxylase activity”. After that, 10 hub genes were identified and their possible relationships with the onset and progression of cancer were evaluated. The authors reported that some transcription factors (i.e., STAT3, NFKB1, FOXC1, HINFP and JUN) also showed correlations with respiratory illnesses and the progression of malignancies. Another bioinformatic analysis reported the upregulation of some tumour-related genes in SARS-CoV-2 patients, particularly among genes that are involved in cell cycle regulation or cellular senescence processes [92].

## 6. Immunosuppression

The immune system plays an important role in the defence against tumour cells. For this reason, the occurrence of tumours is significantly higher in immunosuppressed individuals [93]. Ghosh et al. [94] observed that, unlike other viruses, β-coronaviruses employed lysosomal trafficking for egress rather than the biosynthetic secretory pathway. In addition, SARS-CoV-2 infects but does not replicate [95] in monocytes and monocyte-derived macrophages, which induces host immunoparalysis and promotes COVID-19 progression [95]. Indeed, macrophage infection induces transcriptional programmes for specific M2 macrophages. Cancers usually stimulate the activation of M2 macrophages, which suppresses immune responses and contributes to tumour development. Phetsouphanh et al. [96] studied patients with LC and compared them to age- and gender-matched uninfected controls. They observed that the LC patients had highly activated innate immune cells, lacked naive T and B cells and presented an elevated expression of type I and III interferons that remained persistently high for up to 8 months after infection.

As previously stated, SARS-CoV-2 disrupts epigenetic regulation [97], thereby suppressing the innate antiviral cell response in its host [98,99]. Histone functions are implicated in the controlled access to the genome. Kee et al. [100] found that SARS-CoV-2 mimicked histone proteins. This promoted the impairment of the host’s ability to effectively regulate gene expression and respond to the infection. This strategy could also delay the activation of the innate immune system response that is related to interferon type I and III. This strategy for improving replication and immune evasion has been previously described in other viruses [101] and could have implications in the development of chronic inflammation and cancer.

It is also important to take into account that SARS-CoV-2 seems to escape direct recognition by TLR4, as shown by van der Donk et al. [102], which could explain the inefficient immunity to SARS-CoV-2 during the early stages of infection. Indeed, plasmocytoid dendritic cells (pDCs) seem to sense SARS-CoV-2 via the following two distinct innate immune pathways: the endosomal TLR7 pathway activated through viral RNA, which leads to type I IFN production, and the TLR2 pathway triggered by the recognition of the viral envelope protein, inducing IL-6 production [103]. NK cells from patients with COVID-19 show a dysfunctional status similar to tumor-associated NK cells, and the numbers of circulating NK cells are decreased [104].

Polymorphisms in killer cell immunoglobulin-like receptors (KIRs) also confer differential viral susceptibility and disease severity. Bernal et al. [105] observed the crucial role of NK in the clinical variability of COVID-19 with specific KIR/ligand interactions associated with disease severity. In fact, the KIR2DS4 gene carried the highest risk for severe COVID-19 infection [106].

## 7. Autophagy

SARS-CoV-2 and other viruses (e.g., HCV) have shown the ability to interfere with autophagic mechanisms. Some SARS-CoV-2 proteins can inhibit autophagic flux at different levels. Nsp15 can block the induction of autophagy (reduction in the number of autophagosomes, decrease in LCB3-II and accumulation of p62), ORF7a can reduce the acidity of lysosomes and ORF3a prevents the fusion of autophagosomes and lysosomes, in addition to promoting lysosomal exocytosis. Altered autophagic flux could lead to protein accumulation, oxidative stress, organelle damage and the disruption of cell cycles, among other outcomes. Together, all of these alterations create perfect microenvironments for cancer development [107]. However, SARS-CoV-2 can also use autophagic mechanisms for its own benefit. Through the viral protein ORF8, SARS-CoV-2 can promote the degradation of the major histocompatibility complex I (MHC-I). This is an escape strategy that is used by cancer cells against the immune system [108].

## 8. Is There Something We Have Overlooked?

To further complicate our knowledge of coronaviruses, some authors have reported unexpected cancer remission (lymphoma) in some patients during acute SARS-CoV-2 infection [109,110,111] or tumour reductions (colorectal cancer) [112] during SARS-CoV-2 infection. The reasons for this are not well known. One reason could be that infection with low-pathogenic SARS-CoV-2 could lead to efficient and rapid oncolysis [113]. Another possibility could be that SARS-CoV-2 could trigger an antitumour immune response [109], which has been described in some lymphoma and other infections [114]. However, Pasin et al. [110] observed a recurrence of lymphoma after the patient recovered from COVID-19. This would suggest a close interaction between the COVID-19 infection, inflammation and tumour biology [111].

## 9. Limitations and Perspectives

From an epidemiological point of view, no studies have reported an increase in cancer incidence so far. One reason for this could be the relatively short period of time since COVID-19 first emerged. However, if a link between COVID-19 and cancer development was confirmed, it would have significant impacts on public health [83]. In the same way, studies on SARS survivors have not reported an increase in cancer incidence [115,116], although the number of cases is much more limited compared to SARS-CoV-2.

Potential biases should be also considered. Perhaps most importantly, the COVID-19 pandemic has been associated with a sharp decline in cancer screening [117,118]. So, a national population-based modelling study that analysed the impact of COVID-19 pandemic on cancer delays and survival suggested that significant increases in the number of avoidable cancer deaths could be expected as a result of diagnostic delays [118]. For this reason, policy interventions are needed to mitigate the indirect effects of the COVID-19 pandemic on patients with cancer [118]. Another aspect to consider is that in some patients with LC, some cancer-related symptoms could be attributed to LC, which could contribute to late-stage cancer diagnoses.

Although Table 1 summarises the possible mechanisms that may or may not contribute to oncogenesis, it is essential to continue searching for and updating the arguments for and against this possible association. The expression of some SARS-CoV-2 proteins can have oncogenic effects (Figure 1), but this does not necessarily imply that they promote cancer development. Similar observations have been reported following respiratory syncytial virus infections where M protein expression profoundly affects the cell cycle through a p53-dependent pathway [119]. For all of these reasons, and based on previous experience with infectious diseases, interdisciplinary teams should study LC patients to investigate the relevant aspects. Firstly, we need to monitor LC patients for the appearance of new comorbidities. Secondly, it is essential to identify biomarkers that could allow us to assess the impacts of LC and evaluate possible therapeutic interventions. Thirdly, we need to focus on the development of interventions that aim to control, eliminate and even eradicate these viruses. Fourthly, it would be interesting to study the potential role of SARS-CoV2 vaccination in the pathways that potentially promote cancer. Fifth, should people who have been infected more than once be monitored more closely? Lastly, we need to develop animal models to evaluate the impacts of LC [120]. Until these objectives are met, it is imperative to continue to conduct studies to provide clarity on this and other issues related to LC.

## Figures and Tables

**Figure 1 microorganisms-11-00713-f001:**
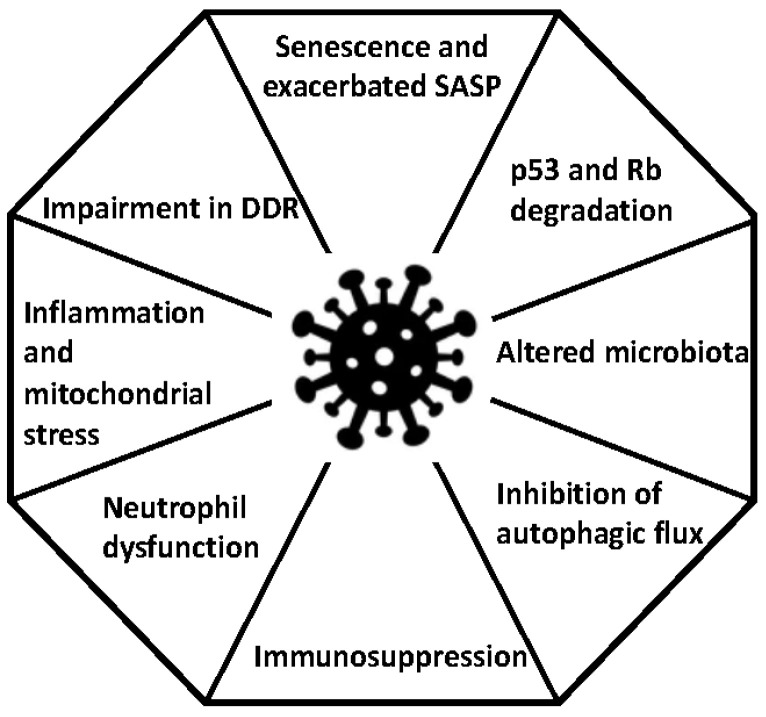
Potential oncogenic mechanisms observed in SARS-CoV-2 infection. DDR = DNA damage response; Rb = retinoblastoma protein; SASP = senescence-associated secretory phenotype.

**Table 1 microorganisms-11-00713-t001:** Arguments that support or argue against the potential implication SARS–CoV–2 in oncogenesis.

In Favour	Against
In senescent and non-senescent cultured human cells SARS-CoV-2 induced senescence and exacerbates the SASP [34,35].	Acceleration of the biological blood clock reversed in some patients [39].
Senescence mechanisms could be mediated by DNA damage and activation of the DNA damage response pathway [36].	No differences on epigenetic age or telomere length [41].
DNA methylation changes [37].	Persons with LC have not significative changes on cytokine levels of IFN-γ, TNF and IL-6 [47].
Acceleration of the biological blood clock [38] and reduction on the telomere length in Vero E6 cells [40].	No detection of viral RNA of patients recovering from COVID-19 [62].
Significant inflammatory response associated wih anti-inflammatory response and mitochondrial stress [45].	The presence of SARS-CoV-2 RNA months after infection does not necessary indicates the presence of live virus [63].
Neutrophil dysfunction [46].	Non-culture of SARS-CoV-2 from samples with persistence of viral antigen [68].
Persons with LC have significant higher levels of IL–17 and IL–2 [47].	No integration into human genome of SARS-CoV-2 [73,74,75].
Residual virus reactivation [49,50].	In-silico study model not evidence of a genetic correlation with any of the female-specific cancers [90].
SARS–CoV–2 promotes inflammasome activation and inflammatory pathways [54,55].	So far, studies on SARS survivors have not reported an increase in cancer incidence [115,116].
Detection of residual SARS-CoV-2 in multiple tissues from patients recovering from COVID-19 [62].	
Presence of SARS-CoV-2 RNA and/or antigens months after infection [63,68,69].	
Indirect evidence suggesting the viral reservoir hypothesis based on the potential benefits of protease inhibitors in persons having LC [71].	
Some tumour suppressor genes could be inhibited by SARS-CoV-2 [79,83,85].	
Cross-trait meta-analysis identified some conditions that share multiple mechanistic pathways with some female cancers [90].	
Bioinformatics studies suggest that some transcriptional factors show a correlation with respiratory illness and progression of malignancies [91].	
Bioinformatic studies show the up-regulation of some tumour–related genes [92].	
SARS–CoV–2 mimics the histone proteins [100].	

DEGs = differentially expressed genes; IL = interleukin; LC = long COVID.

## Data Availability

Not applicable.

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
