# Peer review of "Should We Expect an Increase in the Number of Cancer Cases in People with Long COVID?"

_microorganisms, 2023, doi:10.3390/microorganisms11030713_

Round 1

Reviewer 1 Report

I read with interest the review article “Should we Expect an Increase in the Number of Cancers Cases

Following the SARS‐CoV‐2 Pandemic” by Ana-Amiama Roig et al. In general it is well writes and provides thoughtful insights about the topic.

Minor comments

1. Line 280 “were key were in cancer development”. Please delete the second “were”

2. Conclusions line 371:the development of oncologic complications,”

The meaning is about the emergence or the acceleration of cancer and not of its complications. Please rephrase.

General and Major comments:

A figure would increase the article’s visibility.

There are some important areas that have not been discussed. The authors are kindly asked to add a paragraph for each one:

1.       What would be the potential role of SARS-CoV2 vaccination in the discussed pathways potentially promoting cancer. In other words, patients who get infection after vaccination may have a lower risk as it appears also for long covid?

2.       People with multiple SARS-CoV2 infections would exhibit increased oncogenic risk?

3.       What about those treated with antivirals versus not treated?

4.       Most data about senescence, inflammation and other pathways discussed herein derive from blood assays. Organ-specific pathways might be different, depending on the abundance of ACE2 receptors and the ability of each tissue to clear the virus, its dead components or the activated cascades. Finally repair mechanisms may be less effective in various tissues and in addition to other toxic exposures.

Author Response

Thank you for your comments and suggestions. We have included it in the limitations sections.

Reviewer 2 Report

A well-written, comprehensive review of the literature on the possible relationship between SARS-CoV-2 infection and the risk of cancer.

(Minor) comments:

Line 61: Given the focus on the possible risk of cancer associated with long-COVID in the Section 9, LC should be mentioned here as well. Suggestion: ‘…we reviewed the different arguments that support or reject the possibility of a correlation between acute and long COVID‐19 and the risk of cancer ….’

Line 82: ‘CS’ instead of ‘CD’

Line 82: ‘double-edged’ instead of ‘double edge’

Line 122: ‘conditions’ instead of ‘situations’

Line 150: ‘No change in IL-6’ instead of ‘This’

Section 4: the authors may want to include reference to this recently published paper: Bussani, R., Zentilin, L., Correa, R., Colliva, A., Silvestri, F., Zacchigna, S., Collesi, C. and Giacca, M. (2023), Persistent SARS-CoV-2 infection in patients seemingly recovered from COVID-19. J. Pathol., 259: 254-263. https://doi.org/10.1002/path.6035

Lines 162-167 - References 54 &55 do not describe Spike-mediated or furin cleavage site-mediated activation of the inflammasome. Please, amend sentence or include correct references.

Line 280: delete 2nd ‘were’

Line 373: The authors make the following comment: ‘but this does not necessarily imply that they promote cancer development.’ In line with this comment, Respiratory Syncytial Virus has also been shown to down-regulate p53 and pRb (Bian et al, PLosOne 2012; https://pubmed.ncbi.nlm.nih.gov/22662266/), without increasing the risk for cancer (to my knowledge).

Suggestion for title of Table 1: ‘Arguments that support or argue against the potential implication of SARS-CoV-2 on oncogenesis.’

Author Response

Thank you very much for your comments and suggestions. We have included it.

Reviewer 3 Report

The review by Amiama-Roig et al is a comprehensive compendium of many reported manuscripts regarding the possible role of SARS-CoV-2 on the risk of developing cancer analyzing arguments that could support or reject this fact. It is well written and the bibliography included is appropriate.

I have only minor comments:

* Line 82: CS instead of "CD"

* Table 1 should have more resolution.

*  In section 6 "Immunosuppression", authors should include some studies related to TLR, KIR, and dendritic cells, which are reported to be altered in patients with SARS-CoV-2 or convalescents and have a potential to impair innate immunity.

Author Response

Thank you very much for your comments.

We have included it.
